# Engineering an Effective Human SNAP-23 Cleaving Botulinum Neurotoxin A Variant

**DOI:** 10.3390/toxins12120804

**Published:** 2020-12-18

**Authors:** Stefan Sikorra, Sarah Donald, Mark Elliott, Susan Schwede, Shu-Fen Coker, Adam P. Kupinski, Vineeta Tripathi, Keith Foster, Matthew Beard, Thomas Binz

**Affiliations:** 1Institut für Zellbiochemie, OE 4310, Medizinische Hochschule Hannover, 30623 Hannover, Germany; S.Sikorra@kabelmail.de (S.S.); SusanSchwede@gmx.de (S.S.); 2Ipsen Bioinnovation, 102 Park Drive, Milton Park, Abingdon OX14 4RY, UK; sarah.donald@ipsen.com (S.D.); mark.elliott@ipsen.com (M.E.); shu-fen.coker@ipsen.com (S.-F.C.); adam.kupinski@ipsen.com (A.P.K.); vineeta.tripathi@ipsen.com (V.T.); drkafoster@gmail.com (K.F.)

**Keywords:** SNAP-25, SNAP-23, zinc protease, botulinum toxin, substrate specificity, screening method

## Abstract

Botulinum neurotoxin (BoNT) serotype A inhibits neurotransmitter release by cleaving SNAP-25 and represents an established pharmaceutical for treating medical conditions caused by hyperactivity of cholinergic nerves. Oversecretion from non-neuronal cells is often also the cause of diseases. Notably, excessive release of inflammatory messengers is thought to contribute to diseases such as chronic obstructive pulmonary disease, asthma, diabetes etc. The expansion of its application to these medical conditions is prevented because the major non-neuronal SNAP-25 isoform responsible for exocytosis, SNAP-23, is, in humans, virtually resistant to BoNT/A. Based on previous structural data and mutagenesis studies of SNAP-23 we optimized substrate binding pockets of the enzymatic domain for interaction with SNAP-23. Systematic mutagenesis and rational design yielded the mutations E148Y, K166F, S254A, and G305D, each of which individually increased the activity of LC/A against SNAP-23 between 3- to 23-fold. The assembled quadruple mutant showed approximately 2000-fold increased catalytic activity against human SNAP-23 in in vitro cleavage assays. A comparable increase in activity was recorded for the full-length BoNT/A quadruple mutant tested in cultivated primary neurons transduced with a fluorescently tagged-SNAP-23 encoding gene. Equipped with a suitable targeting domain this quadruple mutant promises to complete successfully tests in cells of the immune system.

## 1. Introduction

The botulinum neurotoxins (BoNTs) are produced by various members of the gram-positive bacterial genus Clostridium and are considered the most hazardous natural substances to humans [1]. However, for 30 years mainly serotype A (BoNT/A), but in some indications also serotype B (BoNT/B), have served as effective pharmaceuticals for the treatment of medical conditions caused by hyperactivity of cholinergic nerve terminals. BoNT/A was initially approved for the treatment of blepharospasm, hemifacial spasm, and strabismus. Since then the range of applications has steadily expanded and now includes a variety of neuromuscular conditions as well as autonomic and other non-neuronal uses [2,3,4].

There are seven serotypes of BoNTs (types A–G) as well as various subtypes within serotypes A, B, E, and F. There are also recently discovered types including serotype X and several non-clostridial BoNTs. All of them are synthesized as ~150 kDa single-chain proteins and subsequently cleaved to yield an enzymatic light chain (LC) of ~50 kDa and a heavy chain of ~100 kDa. The C-terminal half of the heavy chain mediates the highly specific targeting of neuronal cells harnessing polysialogangliosides and synaptic vesicle proteins for binding and receptor-mediated endocytosis [5]. The N-terminal half of the heavy chain (HN) forms a channel subsequent to acidification in endocytic vesicles and delivers the LC to the cytosol. In the cytosol the LCs act as zinc metalloproteases and hydrolyze specific SNAREs (soluble N-ethyl maleimide sensitive factor attachment protein receptors). BoNT/B, D, F, and G cleave synaptobrevin/vesicle associated membrane protein isoform 1, 2, and 3. BoNT/A, C, and E cleave SNAP-25 (synaptosomal-associated protein of 25 kDa) and type C attacks in addition syntaxin 1, 2, and 3. SNARE cleavage prevents neurotransmitter release for a serotype-specific duration, resulting in the characteristic botulism symptoms of a flaccid paralysis that can lead to respiratory failure and death [6,7].

The expansion of the use of BoNTs to further medical conditions that are associated with hypersecretion from non-neuronal cells like the widespread diseases chronic obstructive pulmonary disease, asthma or diabetes is hampered because respective SNAREs are refractory to BoNT LCs and appropriate receptor molecules are lacking on target cells. In order to deliver the BoNT enzymatic domain to cells not targeted naturally by the toxins, technologies like the targeted secretion inhibitor (TSI) platform or SNARE tagging technology have been established [8,9]. Both are based on engineered proteins that incorporate the LC domain and HN translocation domain, the LC plus HN fragment (called LHN), together with a binding domain that binds to a cell surface receptor on the desired target cell. By targeting membrane receptors or cell surface proteins that internalize into the cell, the HN domain is enabled to enter a suitable endosomal compartment in which it can form a pore to enable entry of the LC into the cytosol. Whereas in TSIs LHN and binding domain are directly linked, SNARE tagging connects them by means of attached SNARE domains that form a highly stable tetrahelical complex. The LHN/C and LHN/D based TSIs linked to epidermal growth factor and growth hormone releasing hormone, respectively, have been effective in decreasing release of mucin in a pulmonary cell line and growth hormone in anterior pituitary somatotroph cells, respectively [10,11]. Successful reduction of hormone secretion from a variety of neuroendocrine tumor cell lines has been reported for LHN/A SNARE tagged to ciliary neurotrophic factor, corticotropin-releasing hormone and epidermal growth factor [12].

To overcome the fact that SNAREs mediating secretion in non-neuronal cells are usually refractory to BoNT LCs, specific binding pockets have to be adapted to the proposed SNARE target. So, human SNAP-23 (hSNAP-23), an isoform of the BoNT/A substrate SNAP-25 represents an attractive target. This SNARE is involved in secretion from non-neuronal cells, such as myeloid cells [13,14,15], but was reported to be virtually resistant to SNAP-25 cleaving BoNTs [16,17,18]. Likewise, none of the growing number of characterized LC/A subtypes has been reported to proteolyze hSNAP-23 [19,20,21]. Thus, a step towards a potential therapeutic for hypersecretion diseases of non-neuronal tissue would be to engineer a LC/A variant that efficiently proteolyzes hSNAP-23.

Here we selectively mutated the prototypical LC/A (subtype 1) to improve its proteolytic activity against hSNAP-23. We identified four mutations that increased the activity ~3- to 23-fold. Notably, individual combinations of these mutations led to measurable synergistic effects on proteolytic activity yielding a quadruple mutant that readily cleaved hSNAP-23 in in vitro cleavage assays as well as in neuronal cell-based assays. Further biochemical characterization of this mutant is required but it appears to hold the potential to be advanced to novel pharmaceuticals for the treatment of diseases that rely on SNAP-23 mediated hypersecretion.

## 2. Results

Human SNAP-23 was confirmed to be virtually resistant to proteolysis by LC/A; micromolar LC concentrations caused only minimal cleavage in in vitro cleavage assays (Table 1 and Table 2), whereas its neuronal isoform, SNAP-25, was cleaved effectively by sub-nanomolar concentrations [18,22]. To pinpoint the cause of this difference we determined the enzyme kinetic parameters for hSNAP-23 and SNAP-25 cleavage. hSNAP-23 cleavability was ~120,000-fold lower compared to SNAP-25, in particular due to a more than 5000-fold lower turnover number (0.20 * 1/min vs. 1026 * 1/min). Thus, development of a LC/A-based hSNAP-23 targeting therapeutic requires tremendous improvement of the catalytic activity. To achieve this, we employed two complementary approaches both based on a recently published LC/A-hSNAP-23 structural model. Firstly, we systematically mutated LC/A binding pockets to hSNAP-23 residues mainly responsible for its resistance to cleavage and selected LC/A mutants showing increased cleavage by a yeast–based screening method (Section 2.1). Secondly, we created and tested rationally designed LC/A mutants (Section 2.2). Afterwards, mutations that caused increased hSNAP-23 cleavage activity were assembled in one LC molecule and tested whether combinations of mutations lead to additive effects in proteolytic activity to hSNAP-23 (Section 2.3).

### 2.1. Screening LC/A Libraries for Mutants Exhibiting Increased hSNAP-23 Cleavage Activity

We have previously built a model of the complex of hSNAP-23 bound to LC/A and conducted hSNAP-23 and SNAP-25 mutagenesis analyses. These studies revealed a similar binding mode of hSNAP-23 and SNAP-25 and that, of the 23 amino acids differing in the LC interacting region, exchange of Arg-176 by proline and of Thr-200 by lysine showed most detrimental effects on SNAP-25 cleavage [18]. Thus, to re-target LC/A to hSNAP-23 we first set out to optimize the binding pockets for these two amino acids. Systematic mutation of the binding pockets for the equivalent positions in hSNAP 23 included E-148, T-307, A-308, and Y-312 for Pro-182 and L-256, V-258, L 367, and F-369 for Lys-206. For the identification of LCs showing enhanced hSNAP-23 cleavage, we applied a recently introduced yeast based screening method which allows parallel testing of thousands of mutants [18,23]. Such a method is indispensable when three or more LC residues of a binding pocket are simultaneously systematically mutated in a library. LC/A mutants were expressed under the control of the Gal promoter and the weaker Cyc1 promoter.

Screening experiments led to the isolation of a total of 24 clones for the Pro-182 binding pocket. Mutations identified for these clones are listed in Appendix A. The consensus for each of the four positions is a hydrophobic amino acid. This seems to be plausible as hydrophobic (yellow in Appendix A) or slightly hydrophobic (pale yellow in Appendix A) side chains may form a favorable environment for interactions with the aliphatic part of Pro-182. Amino acids carrying the most hydrophobic side chains (F, I, L, M, V) were particularly enriched. On the basis of plausibility, three of the mutants were selected for biochemical characterization. To this end, mutations were transferred into bacterial expression plasmids separately for position 148 and for positions 307 to 312. Combinations of these two sets of mutations (single mutants at site 148 and triple mutants at sites 307/308/312) led to plasmids encoding quadruple mutants. Thus, this procedure allowed for studies of the activity of LC/A point mutants at position 148 as well as of triple mutants at positions 307/308/312, in addition to the quadruple mutants. All mutants were purified as His6-tagged proteins from *E. coli* lysates and studied in standard in vitro cleavage assays. Quadruple mutants 12 (E148N/T307I/A308P/Y312V), 13 (E148Y/T307F/A308N/Y312L), and 21 (E148Y/T307L/A308T/Y312M) exhibited increases in activity of 173%, 101%, and 123%, whereas their related triple mutants, in which E148 was unmutated, showed 74%, 49%, and −2% increased activity respectively (Appendix A). Most interestingly, the strongest increase in activity in this set was observed for LC/A-E148Y single mutant. This mutation is present in three of the isolated yeast clones including mutants 13 and 21 (Appendix A). As the activity of LC/A-E148Y is much higher (728% increase vs. wild-type LC/A) than the activity of the two related quadruple mutants, combinations of mutations in positions 307, 308, and 312 appear to be counterproductive in the context of the quadruple mutant. With respect to mutant 12, the single mutant LC/A-E148N (12a in Appendix A) caused merely a marginal activity increase, and LC/A-T307I/A308P/Y312V led just to 74% increase (Appendix A). This suggests a synergistic effect of these mutations in mutant 12.

Activity studies of LC/A-E148Y (12a in Appendix A) using lower concentrations confirmed previous results (Table 1). Determination of enzyme kinetic parameters showed that its increased cleavage of hSNAP-23 is mainly due to a 4.5-fold reduction of Km. However, an almost 2-fold increased kcat contributed to the improved activity, too (Table 3).

Screening of mutations in the Lys-206 binding pocket (L256, V258, L367, and F369 of LC/A), led to a total of 20 clones that were characterized by sequencing. Mutated starting amino acids of this binding pocket are space requiring and strongly hydrophobic and thus don’t seem favorable to accommodate the side chain of hSNAP-23 Lys-206. Expectably, we observed an accumulation of amino acids carrying small side chains substituted into positions 258, 367, and 369. Seven, eleven, and seven of them were counted, respectively, when including Gly, Ala, Cys and Ser as amino acids with small side chains (Appendix A). This exceeds random presence by more than factor two. In addition, an enrichment of acidic residues in position 258 was found (Appendix A). This might allow interaction with the positively charged amino group of Lys-206. We chose mutants 9 (L256D/V258S/L367A/F369S) and 10 (L256D/V258A/F369G) for further characterization, since they comprise each one amino acid with carboxylate group and two with small side group. We generated first double mutants comprising position 256 and 258 or 367 and 369 separately before combining to create a quadruple mutant. Neither the four double mutants nor the quadruple mutant 9 (Appendix A) showed increased activity to hSNAP-23 as recombinant His6-tagged LCs isolated from *E. coli* lysates. As protein yields of these mutants were quite low, we assume these mutations caused detrimental effects on structural integrity of LC/A which precluded determination of their realistic activity in the applied in vitro assay method, unlike their activity in live yeast.

### 2.2. Rational Design of hSNAP-23 Cleaving LC/A Mutants

A second approach to increase LC/A activity to hSNAP-23 was rational design. The structural model of hSNAP-23-bound LC/A was surveyed, aiming at identifying opportunities for mutations that would establish additional sites of interaction between LC/A and hSNAP-23. As the Km value of LC/A interaction with hSNAP-23 is much higher than that for SNAP-25 [18,22] it seemed reasonable that increasing the affinity of substrate binding would be a good strategy to increase cleavage activity. Based on crystal structure data and results from molecular dynamic simulation studies, we selected LC/A binding pocket residues for hSNAP-23 amino acids where the interactions appeared sub-optimal. These included Lys-185 (Asp-179 in SNAP-25), Arg-186 (conserved position, Arg-180 in SNAP-25), Ile-198 and or Ile-200 (Ile-192/Glu-194 in SNAP-25), and Asp-210 (Gly-204 in SNAP-25). LC/A residues V-304 and G-305 were each mutated to aspartic acid, glutamic acid, and glutamine. These exchanges might allow salt-bridge or H-bond formation with hSNAP-23 Lys-185 and thereby stabilize the interaction (Figure 1D). In addition, these mutations, which are all to acidic residues, are expected to be unfavorable for interaction with the SNAP-25 counterpart amino acid at this position, Asp-179. The same approach was applied to LC/A S-143 in order to affect an interaction with Arg-186 of hSNAP-23 (conserved position, Arg-180 in SNAP-25). Furthermore, LC/A K-166 was mutated to various amino acids carrying hydrophobic substituents to allow for interactions with hSNAP-23 Ile-198 and or Ile-200 (Figure 1C; Ile-192/Glu-194 in SNAP-25). Finally, we tried to increase LC/A activity to hSNAP-23 by disabling the proposed interaction between LC/A-S-254 with hSNAP-23-Asp-210 (Figure 1E; the residue at this position in SNAP-25 is Gly-204) at the so-called β-exosite [24]. Disrupting this interaction was considered potentially beneficial as it might generate a better leaving group upon cleavage and thus increase the turnover number of LC/A. Activity of recombinant mutant LCs was tested in in vitro cleavage assays. Table 1 and Table 2 show single, double, triple, and quadruple mutations lead to significantly increased proteolysis of hSNAP-23. Strongest effects were detected for mutations of K-166. LC/A-K166F readily cleaves hSNAP-23 at 100 nM final concentration, whereas wild-type hardly cleaves at 3 µM final concentration (Table 1). Mutations at positions 143, 304, 305, and 254 increased LC/A activity on average 2- to 3-fold, except that S254R showed no increase in activity (Table 1 and Table 2).

As mutated amino acids in positions 304/305 and in position 143 target neighboring substrate residues we next studied if there would be compatibility among those mutations. Neither of the double mutants tested showed improved activity versus corresponding single mutants (Table 2). Next, we studied the effect on enzymatic parameters of mutations which lend themselves most suitably to assembling multiplex mutants. In agreement with the fact that D-305 is located remote to the catalytic center LC/A-G305D exhibits a much reduced Km value, without significantly altering kcat (Table 3). The K166F exchange affects Km and kcat both by roughly a factor of 4.5. This is probably due to the position of this residue at the opening of the active site cavity. Interaction at this region appears to be very important as this mutation increases the catalytic efficiency of LC/A to hSNAP-23 by a factor of 23.

### 2.3. Assembly of hSNAP-23 Adapted LC/A Multiplex Mutants and Their Biochemical Characterization

Of the mutations destined for assembly in multiplex LC/A mutants we first generated the three double mutants arising from E148Y, K166F, and G305D, to test whether individual effects were additive. Each double mutant showed increased activity in the standard in vitro cleavage assay compared to their constituent mutants (Table 1). Their increased activity was approximately proportional to the activity of the constituent single mutants. Notably, LC/A E148Y/K166F readily cleaved hSNAP-23 at 30 nM final concentration. Therefore, we next constructed the corresponding triple mutant LC/A-E148Y/K166F/G305D whose activity again exceeded that of any double mutant (Table 1). In addition, we examined the S254A mutation in the context of the most active double mutant LC/A-E148Y/K166F. Incorporation of S254A also led to a further increase of proteolytic activity (Table 1). Thus, we finally constructed a quadruple mutant comprising all afore described beneficial mutations, LC/A-E148Y/K166F/S254A/G305D (termed LC/A-quadruple). LC/A-quadruple exhibited significantly increased activity compared to both analyzed triple mutants, indicating that none of the mutations negatively influence one another (Table 1).

The various multiple mutants were characterized in more detail by determining their enzyme kinetic parameters in the standard cleavage assay by varying the substrate concentration. LC/A-E148Y/K166F exhibited almost 3-fold higher affinity and a 4- to 9-fold higher kcat compared to the respective single mutants (Table 3). Additional presence of G305D surprisingly increased kcat but not Km. The reason for this effect is currently unknown. Incorporation of S254A in LC/A-E148Y/K166F expectably had no effect on Km, but increased kcat roughly 4-fold (see above, Table 3). Compared to both triple mutants LC/A-quadruple exhibits a slight increase in affinity and about a 2-fold increase in kcat (Table 3). When comparing the kinetic parameters of LC/A quadruple for hSNAP-23 with those of LC/A wild-type for the authentic substrate SNAP-25, Km values are similar, but kcat is still a factor of 37 higher for LC/A wild-type. In conclusion, merging individually designed single mutants in one LC/A molecule led to an enzyme that cleaves the previously largely refractory SNARE hSNAP-23 with 2000-fold higher catalytic efficiency. However, compared to the activity of LC/A wild-type on SNAP-25 its catalytic efficiency is still a factor of 60 lower (Table 3).

### 2.4. Activity of hSNAP-23 Adapted LC/A Mutants on SNAP-25

One desired characteristic of an LC/A based therapeutic cleaving hSNAP-23 is lacking or reduced activity on the authentic substrate. Therefore, the effects of the individual mutations assembled in the quadruple mutant on SNAP-25 cleavage were studied using standard cleavage assays. K166F had no significant effect on substrate cleavage efficiency, whereas E148Y led to about 50% reduction (Appendix A). This is plausible, because structural studies predicted a strong electrostatic interaction of E148 carboxylate with the Arg-176 guanidinium group of SNAP-25, whereas K166 forms (if any) hydrophobic interactions via its methylene groups with SNAP-25 Ile-192. Since the mutation S254A likely helps generate a better leaving group through abolishing an interaction with the C-terminal substrate cleavage product, it was to be expected that LC/A-S254A showed an increase in activity against SNAP-25 as well as to hSNAP-23. This amounted to ~60% for SNAP-25 cleavage, which was far less pronounced compared to the effect on hSNAP-23 cleavage (Table 1; Appendix A). The effect of the G305D mutation was studied in the context of E148Y and K166F mutations. G305D reduced the negative effect on SNAP-25 cleavage observed for E148Y in E148Y/G305D and E148Y/K166F/G305D, suggesting that G305D also increases LC/A activity on SNAP-25 (Appendix A).

Overall, of the mutations assembled in LC/A quadruple, only E148Y led to the second desired effect, a decrease in SNAP-25 cleavage activity. Thus, a follow-up study should take up this issue and identify LC/A residues for mutation which are exclusively involved in interaction with SNAP-25.

### 2.5. Substrate Cleavage of BoNT/A Quadruple Mutant in Cortical Neurons

Though LC/A quadruple is more than an order of magnitude less active in cleaving hSNAP-23 compared to wild-type cleavage of SNAP-25 in biochemical assays, we next inspected whether its in vitro activity would translate into hSNAP-23 cleavage activity in a cell-based assay. To this end we produced and purified full-length BoNT/A quadruple. Rat cortical neurons were chosen as a test system because they express the extracellular receptors for BoNT binding and uptake. As these cells do not naturally express SNAP-23 they were transduced with Lentivirus expressing either an hSNAP-23-GFP or hSNAP-23-mCherry fusion protein at day six in vitro (DIV 6). At DIV 18 BoNT/A and BoNT/A-quadruple were administered to wild-type and virally transduced cells at varying concentrations. After 24 h of incubation cells were lysed and lysates analyzed for cleavage of hSNAP-23 and the endogenously expressed SNAP-25 by Western blotting. hSNAP-23 was detected using a rat polyclonal antiserum to hSNAP-23. Uncleaved hSNAP23-GFP, or hSNAP-23-mCherry, migrated at about 55 kDa, whereas the presumed hSNAP-23 cleavage product ran as a 25 kDa band (Figure 2A). The arrowhead marks an uncharacterized product observed in cells transduced with hSNAP-23-mCherry with or without BoNT intoxication. Endogenously expressed SNAP-25 was analyzed as control using an antibody to the N-terminus. The motility of the cleavage product is slightly faster due to loss of nine C-terminal amino acid residues (Figure 2B). Quantification of several experiments revealed that recombinant BoNT/A-quadruple and recombinant BoNT/A wild-type cleave endogenous SNAP-25 with equal efficiency, demonstrated by EC_50_ values of 2.87 ± 0.60 and 1.59 ± 0.17 pM (mean ± s.e.m) respectively, in assays conducted with hSNAP-23-mCherry (Figure 2C) or 0.91 ± 0.28 and 1.59 ± 0.17 pM (mean ± s.e.m) in assays with hSNAP23-GFP (Appendix A). There was no statistically significant difference in SNAP-25 cleavage by recombinant BoNT/A-quadruple and recombinant BoNT/A wild-type as measured by unpaired t-test (*p* > 0.05, see Appendix A). In agreement with the reported inactivity of LC/A on hSNAP-23 [17,18], treatment of neurons with wild-type BoNT/A did not result in detectable cleavage of either hSNAP-23-mCherry or hSNAP-23-GFP (Figure 2C, Appendix A). In contrast, BoNT/A-quadruple proteolyzed hSNAP-23 readily. In hSNAP-23-mCherry-based experiments the EC_50_ value of cleavage was determined as 512 pM (mean of four experiments). Thus, the activity of BoNT/A-quadruple on hSNAP-23 is about factor 322 lower compared to the activity of wild-type on SNAP-25 (taken as a ratio of mean EC_50_ for SNAP-23 over the mean EC_50_ for SNAP-25). This is roughly in agreement with the 60-fold lower activity observed in in vitro cleavage assays using recombinant substrates.

## 3. Discussion

BoNT/A is an established pharmaceutical for research and treatment in neuromuscular conditions and has also been successful in autonomic and non-cholinergic uses [2,3,4]. The expansion of its use to medical conditions of non-neuronal nature that are associated with hypersecretion components, like hypersecretion from myeloid cells, is prevented by the resistance of the relevant SNARE, hSNAP-23 [17,18]. Therefore, our goal was to engineer LC/A variants that cleave hSNAP-23 at physiologically relevant concentrations.

Previous work provided a structural model for hSNAP-23 interaction with LC/A as well as substrate residues that are responsible for the very poor cleavability of hSNAP-23, the two most import residues being Pro-182 and Lys-206 [18]. Consequently, we first addressed the corresponding LC/A binding pockets and did systematic mutagenesis of the four LC/A residues supposed to form each of these pockets. As systematic mutation at four positions in parallel allows for 160,000 different mutants, a previously established yeast-based screening system [18,23] was used to isolate LC/A mutants that efficiently cleave hSNAP-23. Concerning the Pro-182 binding pocket the mutation E-148 to tyrosine was found to improve LC/A activity to hSNAP-23 8-fold (Table 3) primarily by > 4-fold reduction of Km. Structural inspection revealed that this increased affinity is most likely induced through an H-bond interaction between Y-148 hydroxyl-group and the carboxylate of hSNAP-23 Asp-172, these residues being located at a distance of 2.2 Å. A potentially hydrophobic interaction with Pro-176 side group is unlikely, as this is about 5.2 Å away from the Y-148 phenyl ring. Mutant 12, which contained the substitutions E148N/T307I/A308P/Y312V (Appendix A) showed 2.7-fold increase in activity, albeit that the single mutation asparagine in position 148 showed no significant beneficial effect on hSNAP-23 cleavage (Appendix A). Consequently, isoleucine, proline or valine in positions 307, 308, and 312, respectively, appear to improve interaction with hSNAP-23. The respective triple mutant (mutant 12b; Appendix A), however, led to only 74% increased activity. Nevertheless, it might be worth testing these mutations individually in combinations with E148Y with prospect of further increases in hSNAP-23 cleavage activity. Regarding the Lys-206 binding pocket, 20 clones were isolated that led to hSNAP-23 cleavage in yeast cells. These clones showed a clear preference for amino acids with short side groups at least for positions 258 and 367 as well as for an acidic side group in position 258 (Appendix A). This was plausible, as such mutations would be expected to both compensate for the required space for lysine in hSNAP-23 compared to threonine in SNAP-25 and allow possible ionic interaction with the Lys-206 ε-amine. However, activity tests with His6-tagged mutants 9, 10, and 15, containing these mutations (Appendix A), purified from *E. coli*, did not show increased hSNAP-23 activity. As we observed unusually low yields of these mutants, when expressed as recombinant proteins in *E. coli*, we assume that mutation in this binding pocket interferes with structural integrity of LC/A and thus disallows to optimize this binding pocket for interaction with hSNAP-23 Lys-206.

Rational design of LC/A with increased activity to hSNAP-23 carried out in parallel yielded amongst others G305D and V304D, both being more than 2-fold more active than wild-type. This is likely due to increased affinity as shown for LC/A-G305D whose Km is a factor of four lower. In agreement with this, both D-305 and D-304 are likely able to form H-bond interactions with the amino group of hSNAP-23-Lys-185, which is located according to our structural model at a distance of 3.0 and 3.2 Å, respectively [18]. The highly beneficial effect of replacing LC/A-K166 by hydrophobic amino acids is difficult to explain. The four most hydrophobic amino acids were tested and each one improved hSNAP-23 cleavage. Valine which is regarded the least hydrophobic amino acid in published hydrophobicity scales showed the least effect (Table 1). However, isoleucine, phenylalanine, and leucine display very similar hydrophobicity but isoleucine caused a comparable effect to valine, whereas leucine caused a much stronger improvement of hSNAP-23 cleavage, and phenylalanine led to a tremendous gain in activity (Table 1; a 23-fold increase in catalytic efficiency (Table 3)). The structural model suggests a hydrophobic interaction of the aliphatic part of K-166 with hSNAP-23-Ile-198 (Appendix A in ref. [18]). Our working hypothesis is that the interaction mediated by a phenyl group is much stronger, because it allows an additional interaction with hSNAP-23-Ile-200 which is also located in close proximity (Figure 1C). This might also apply when K-166 is replaced by leucine. Despite these considerations mere hydrophobic interactions are unlikely to explain the observed strong increase in catalytic activity. Therefore, we also assume that this residue occupies a crucial position in the substrate binding channel, at the entrance to the enzyme’s active site, where it can act as an anchor point that supports presentation of the cleavage site residues in a favorable conformation to the active site residues. The K166F mutation showed no effect on SNAP-25 cleavage (Appendix A), which might be due to the fact that hSNAP-23-Ile-200 is replaced by glutamic acid in SNAP-25. Mutation of LC/A-S-254 to alanine increased activity by increasing kcat by about 4-fold. This is compatible with the idea of more efficient release of the C-terminal cleavage product, if the proposed interaction of serine’s hydroxyl group is ablated. Our structural model indicates an interaction with Asp-210 via the carboxyl group (6.4 Å distance) or its carbonyl oxygen (3.4 Å distance; Figure 1E). The latter appears more likely as the mutation also improves SNAP-25 cleavage in which the aspartic acid residue is replaced with glycine.

The assembly of four of the most beneficial mutations led to LC/A-quadruple which cleaved the largely refractory hSNAP-23 at low nanomolar concentrations in in vitro cleavage assays but was still about factor 60 less efficient compared to SNAP-25 cleavage by LC/A wild-type (Table 3). Similar results were previously obtained for LC/E single mutant K224D [25], but BoNT/E is not used as pharmaceutical due to the short half-life of its effect. The difference in hSNAP-23 versus SNAP-25 cleavage efficiency for full-length BoNT/A-quadruple and wild-type BoNT/A, respectively, in cortical neurons was somewhat more pronounced, a factor of 275 in hSNAP-23-mCherry expressing neurons. Intracellular localization of hSNAP-23-mCherry, which has not been analyzed yet, is an aspect that possibly explains this difference. In addition, the confidence intervals of determined EC_50_ values are rather wide (Appendix A). Future studies will need to examine whether the proteolytic activity of LC/A-quadruple is sufficient to significantly reduce hSNAP-23 mediated exocytosis.

The highly similar activities of full-length BoNT/A-quadruple and BoNT wild-type on SNAP-25 in cortical neurons excludes possible disadvantageous effects of the mutations on LC translocation across the vesicular membrane following receptor mediated entry into target cells. This was a theoretical risk since LC interacts with the translocation domain via a region that is also involved in substrate binding [26]. Another critical aspect that has yet to be studied is whether mutations present in LC/A-quadruple affect the duration of activity. This question will need to be answered before LC/A-quadruple can be considered to enter the next stage of developing a novel pharmaceutical, the interlinkage of LC/A quadruple/HNA with an appropriate cell targeting domain. In conclusion, the LC/A quadruple mutant described in the current study has the potential to form the basis for new molecules intended for therapeutic use in diseases where SNAP-23 mediated hypersecretion is a driver of pathology. However, further investigation and optimization of the properties of the LC/A quadruple mutant in the context of a TSI molecule that can selectively target and deliver the modified LC/A into an appropriate cell type will be required before the full potential can be realized.

## 4. Materials and Methods

### 4.1. Plasmid Constructions

The open reading frame of full-length human SNAP-23 [17] was cloned into the in vitro transcription vector pSP73 (Promega, Mannheim, Germany) and the *E. coli* expression vector pBR-IBA (IBA, Göttingen, Germany) to yield phSNAP-23his6 and pS3-hSNAP-23his6, respectively. Construction of plasmids used for synthesis of C-terminally His6-tagged full-length SNAP-25 in *E. coli* (pBN10) [27] and by in vitro transcription/translation (pSNAP-25his6) has been detailed previously [22]. Mutants of pBN3 encoding LC of subtype 1 (strain 62A, aa 1-448) [22] were generated by PCR applying the GeneTailorTM site-directed mutagenesis system (Life Technologies, Darmstadt, Germany) and suitable primers. All mutations were verified by sequencing. The full-length BoNT/A quadruple mutant encoding plasmid was generated by replacing the LC encoding segment of pH6tBoNTAs-thro by the corresponding segment of the LC quadruple mutant expression plasmid, pBN3. pH6tBoNTAs-thro is a derivative of pBoNTAs [28] in which a His6-tag followed by thrombin recognition sequence (MRGSHHHHHHGSLVPRGS) precedes the N-terminal proline and in which the thrombin recognition site, LVPRGS, was in addition inserted between LC and HN (between S-441 and K-448) replacing the authentic amino acid segment LDKGYN.

A yeast optimized coding sequence for LC/A (yLC/A, amino acids 1-425) was inserted into the pGal-STE2 vector [23] using the EcoRI and XhoI sites, resulting in pGal-STE2-yLCA. STE-yLCA encoding fragment was also inserted into p413CYC1 (ATCC^®^ 87378™) to yield pCyc1-STE2-yLCA using the SmaI and SalI sites in the vector and the STE2-yLCA fragment from pGal-STE2-yLCA obtained by NotI cleavage followed by mung bean nuclease treatment and subsequent XhoI cleavage. The C-terminal part of the hSNAP-23 (encoding amino acids 170–211) open reading frame was subcloned in p424GPD (ATCC^®^ 87357™) using the vector NheI and PstI restriction sites and SpeI/PstI cleaved inserts. For LC/A mutant library constructions, the method of successive PCR was used as detailed previously [18].

### 4.2. Lentivirus Generation

Lentiviral vectors encoding either hSNAP-23-eGFP or hSNAP-23-mCherry fusion proteins were generated by inserting genes between XbaI and BamHI sites in the pCDH-EF1a-MCS lentivector (SBI). HEK293TN cells (SBI), cultured in Dulbecco’s Modified Eagle Medium (DMEM) with 10% fetal bovine serum (FBS) in a humidified 5% CO2 atmosphere at 37 °C, were grown to 80–90% confluency in a T80 cell culture flask. The lentivector plasmids pCDH-EF1-SNAP23a-GS5-eGFP or pCDH-EF1-SNAP23a-mCherry (2 µg) were co-transfected with pPACKH1 packaging plasmids (10 µg) (Systems Biosciences) using Lipofectamine 3000 (Invitrogen). Lentiviral particles in the culture media were collected at 24, 48 and 72 h with media replenished after each collection. Collected media was mixed 5:1 with PEG-IT solution (System Biosciences) to precipitate lentiviral particles and stored at 4 °C until concentration. Lentiviral particles were concentrated by centrifugation at 8000× *g* for 30 min at 4 °C in Lynx 6000 centrifuge using a [F14-14 × 50cy] rotor (Thermo Scientific). Lentiviral pellets were re-suspended in Optimem (Gibco) and stored in aliquots at −80 °C until use.

Lentiviral titers were determined by RT-PCR using a method based on that described by Kutner et al. [29]: Briefly HT1080 cells were transduced with lentivirus and lysed 72 h post-transduction. Genomic DNA was extracted using a PureLink Genomic DNA Kit (Life Technologies) according to manufacturer’s instructions. Woodchuck Hepatitis Virus Post-transcriptional Regulatory Element (WPRE) (a component of the lentiviral vector) and RNaseP genes were amplified using TaqMan master mix in a 7500 Fast RT-PCR system (Thermo Scientific). Lentivector copy numbers in HT1080 cells were normalized to RNaseP gene copies using lentivector (pCDH) and human genomic DNA (Sigma) standard curves to calculate viral titers.

### 4.3. Production and Purification of Recombinant SNAP-25, hSNAP-23, LC/A, and Full-Length BoNT/A

The *E. coli* strain M15[pREP4] (Qiagen, Hilden, Germany) was transfected with pBN10 [27] encoding SNAP-25his6, pBN3 or mutants thereof encoding wild-type or mutated LC/A, or pH6tBoNTAs-thro encoding full-length BoNT/A-quadruple. The plasmid pS3-hSNAP-23his6 encoding hSNAP-23 carrying an N-terminally fused strep-tag and a C-terminally fused His6-tag was transfected into the *E. coli* strain BL21-DE3 (Stratagene Europe, Ebsdorfergrund, Germany). Recombinant proteins were produced during 15 h of induction at 21 °C and purified on Ni^2+^-nitrilotriacetic acid-agarose beads (hSNAP-23 and full-length BoNT/A in addition on Strep-Tactin sepharose beads) according to the manufacturers’ instructions. Full-length BoNT/A and BoNT/A-quadruple mutant were produced under biosafety level 2 containment (Bezirksregierung Hannover, project number A/Z 40654/3/57). Fractions containing the desired proteins were dialyzed against toxin assay buffer (150 mM potassium glutamate, 10 mM HEPES-KOH, pH 7.2), frozen in liquid nitrogen, and kept at −70 °C. The buffer used for recombinant hSNAP-23 was supplemented with 10 mM β-mercaptoethanol (β-ME). Protein concentrations were determined following SDS-PAGE analysis and Coomassie blue staining by means of the LAS-3000 imaging system (Fuji Photo Film, Co., Ltd.) and AIDA 3.51 or Multi Gauge 3.0 software using various known concentrations of bovine serum albumin run as standard. SNAP-25his6 and hSNAP-23his6 were additionally generated by in vitro transcription/translation, using the above described plasmids, the SP6/T7 coupled TNT reticulocyte lysate system (Promega), and [35S]methionine (370 kBq/µL, >37 TBq/mmol, Hartmann Analytic, Braunschweig, Germany) according to the manufacturer’s instructions.

### 4.4. Endopeptidase Assays

Standard cleavage assays contained S3-hSNAP-23his6 or SNAP-25his6 at 20 µM final concentration each plus 1 µL of the transcription/translation mixture of [35S]-methionine-labeled S3-SNAP-23his6 or hSNAP-25h6, respectively, and were conducted in toxin assay buffer (150 mM potassium glutamate, 10 mM HEPES-KOH, pH 7.2). Assays with hSNAP-23 were in addition supplemented with 10 mM β-ME to avoid hSNAP-23 precipitation. Applied final LC/A concentrations are specified in legends to tables. Assays were conducted in a total volume of 10 µL and incubated for 60 min at 37 °C. Reactions were stopped by the addition of 10 µL of prechilled double-concentrated sample buffer [120 mM Tris-HCl, pH 6.75, 10% (*v*/*v*) β-ME, 4% (*w*/*v*) SDS, 20% (*w*/*v*) glycerol, 0.014% (*w*/*v*) bromphenol blue]. Samples were incubated at 37 °C for 30 min and then subjected to SDS-PAGE using 15% tris/glycine gels (acrylamide/bis-acrylamide in 73.5:1 ratio). Subsequently, gels were dried and radiolabeled proteins were visualized employing a FLA-9000 phosphorimager (Fuji Photo Film, Co., Ltd., Tokyo, Japan). Quantification of radiolabeled proteins and their cleavage products was done with AIDA 3.51 or Multi Gauge 3.2 software (Fuji Photo Film) and used to calculate the percentage of substrate cleavage.

For the determination of the enzyme kinetic parameters of selected LC/A mutants, the substrate concentration of the standard assay was varied between 1 to 100 µM in toxin assay buffer (150 mM potassium glutamate, 10 mM HEPES-KOH, pH 7.2, supplemented with 10 mM β-ME). Incubation was done in a final volume of 25 µL of toxin assay buffer. After 2 and 4 min of incubation at 37 °C, aliquots of 10 µL were taken and the enzymatic reaction was stopped by mixing with 10 µL of prechilled double concentrated SDS-PAGE sample buffer. Percentage of cleavage was used to calculate the initial velocity of substrate cleavage. Km and Vmax values were determined by non-linear regression using the GraphPad Prism 4.03 program (GraphPad Software Inc, San Diego, CA, USA).

Several mutants were analyzed in a simplified cleavage assay which essentially follows the standard cleavage assay. In this assay, merely 1 µL of transcription/translation mixture of [35S]-methionine-labeled hSNAP-23his6 served as substrate, and cleavage assays were not supplemented with β-ME.

### 4.5. Yeast-Based Screening of LC/A Mutants Libraries

A yeast strain lacking the capacity to synthesize tryptophan, histidine, and leucine represents the basis of the screening system. Transfection of a plasmid leading to expression of a plasma membrane anchored hSNAP-23-LexA fusion protein enables yeast to synthesize tryptophan. Additional transfection of a plasmid encoding a plasma membrane anchored version of LC/A allows yeast to synthesize histidine. However, growth of yeast colonies occurs only if LC/A liberates the transcription factor LexA through SNAP-23 cleavage. LexA turns on transcription of a gene required for leucine synthesis [18]. Both plasmids were co-transfected into yeast strain MYA423 (ATCC, MATalpha ura3 his3 trp1 LexA((op(x6)-LEU2) by using the standard lithium acetate method. Yeast cells were then plated onto selective plates containing 2% galactose and 1% raffinose that lacked tryptophan, histidine and leucine. Plates were incubated for 5 days at 30 °C. Colonies obtained from screening experiments were picked and incubated overnight at 30 °C in 2 mL YPD medium. Genomic DNA was isolated as described [30]. The LC/A encoding sequence was amplified using a STE2 specific primer and T3 primer and sequenced using suitable LC/A specific primers.

### 4.6. Cell Culture

CTX were prepared from embryonic day 17–18 (E17–E18) Sprague-Dawley rat embryos, as described previously [31]. Briefly, dissected cerebral cortical tissue was collected into ice-cold Hank’s Balanced Salt Solution without Ca^2+^ or Mg^2+^, and then dissociated in papain solution for 40 min at 37 °C following the manufacturer’s instructions (Worthington Biochemical, Lakewood, US). Cortical cells were plated on poly-L-ornithine coated 96-well plates at a density of 20,000 cells/well in 125 µL Neurobasal medium containing 2% B27 supplement, 0.5 mM glutaMax, 1% FBS and 100 U/mL penicillin/streptomycin. Cells were maintained in a humidified atmosphere containing 5% CO_2_ at 37 °C. A further 125 µL Neurobasal medium containing 2% B27, 0.5 mM glutamax was added on DIV 4. Cells were maintained by replacement of half the medium twice per week. On DIV 11, 1.5 µM cytosine β-D-arabinofuranoside was added to the medium to prevent proliferation of non-neuronal cells.

### 4.7. Cell Based SNARE Cleavage Assay

Primary rat cortical neurons were transduced with a lentiviral vector expressing either hSNAP-23-mCherry or hSNAP-23-GFP fusion protein. Lentiviral particles were diluted in cell culture media at a MOI of 2.5 (typically 2 µL/ 250 µL culture media per well) and applied to cells at DIV6. Cells were incubated for 24 h in standard culture conditions (37·°C, 5% CO_2_) after which time, media containing lentiviral particles was removed and replaced with fresh culture media. At DIV 18 wild-type and transduced cells were treated with recombinant full-length wild-type BoNT/A or quadruple mutant employing concentrations varying between 0.1 pM and 10 nM. Cells were inspected visually, under a fluorescent microscope, to confirm mCherry or GFP expression after lentivirus transduction prior to BoNT/A treatment. Successful lentiviral transduction was shown also by Western blotting with anti-mCherry and GFP primary antibodies. After 24 h, cells were lysed by removing all medium and adding sample buffer (25% NuPAGE buffer [Life Technologies, Fisher Scientific] supplemented with 10 mM dithiothreitol and 250 units/µL Benzonase [Sigma]). Lysate proteins were separated by SDS-PAGE and transferred to nitrocellulose membranes. Primary antibodies used were against SNAP-25 (Sigma S9684, Sigma), SNAP-23 (Abcam #ab4114). The secondary antibodies were HRP-conjugated ant-rabbit IgG (Sigma A6154). Proteins were visualized using an enhanced chemiluminescent detection system (Fisher Scientific). Luminescence detection was carried out using a Syngene GeneGnome and image analysis was performed using GeneTools software (Syngene Bioimaging, Cambridge UK). SNAP-25 and SNAP-23 cleavage were monitored by measuring the disappearance of the specific full-length SNARE protein and the appearance of the cleaved fragment of the SNARE protein. The amount of cleaved SNARE protein was expressed as a percentage of the sum of full-length SNARE protein and cleaved product when available. Data was fitted to a 4-parameter logistic equation (upper and bottom asymptotes were constrained at 0 and 100% where appropriate), pEC_50_ and EC_50_ were calculated, R2 for all curves was > 0.98.

## 5. Patents

A patent derived from the work reported in this manuscript is filed as WO 2019/145577 A1.

## Figures and Tables

**Figure 1 toxins-12-00804-f001:**
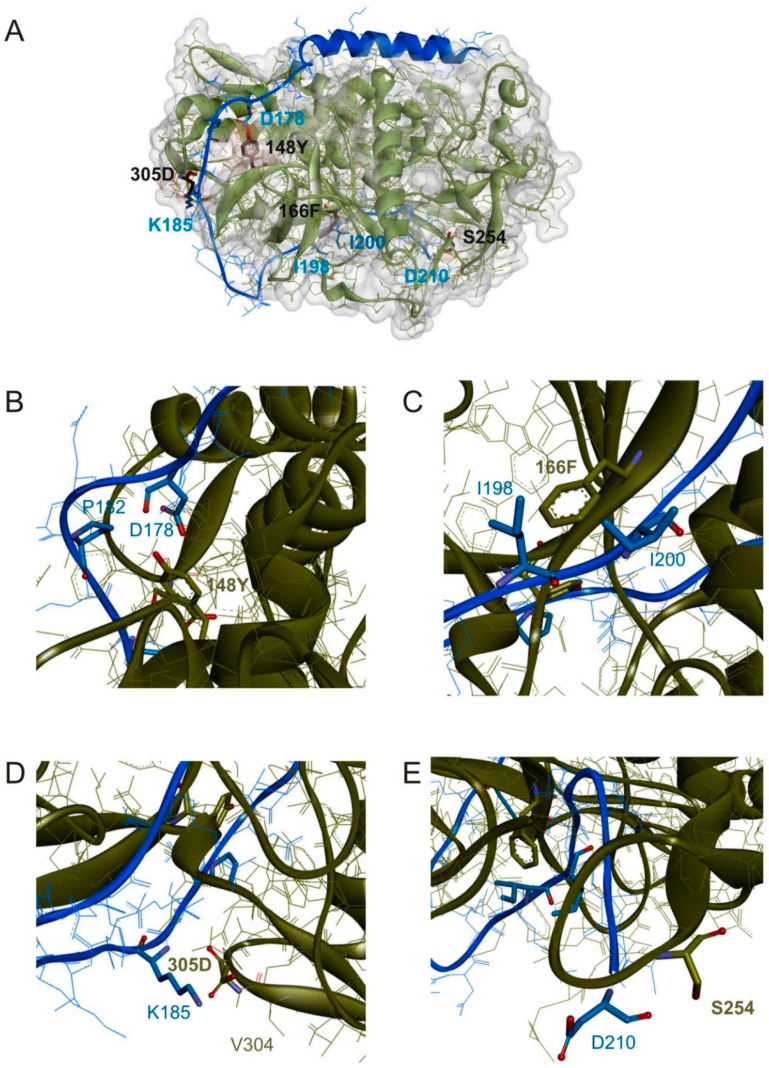
(**A**) Ribbon diagram of a structural model of hSNAP-23 (blue) bound to LC/A (taupe) based on the coordinates of a SNAP-25-LC/A cocrystal structure (PDB ID: 1XTG). LC/A mutations resulting in increased activity to hSNAP-23 assembled in a quadruple mutant are depicted in stick model as well as hSNAP-23 residues presumably interacting with mutated LC/A amino acids. (**B**–**E**) Close-up views of LC/A binding pockets for hSNAP-23 Pro-182 (**B**), Ile-198 (**C**), Lys-185 (**D**), and Asp-210 (**E**). LC/A is colored brown, whereas hSNAP 23 is kept blue. Mutated LC/A residues and putatively interacting hSNAP-23 residues are shown in stick model.

**Figure 2 toxins-12-00804-f002:**
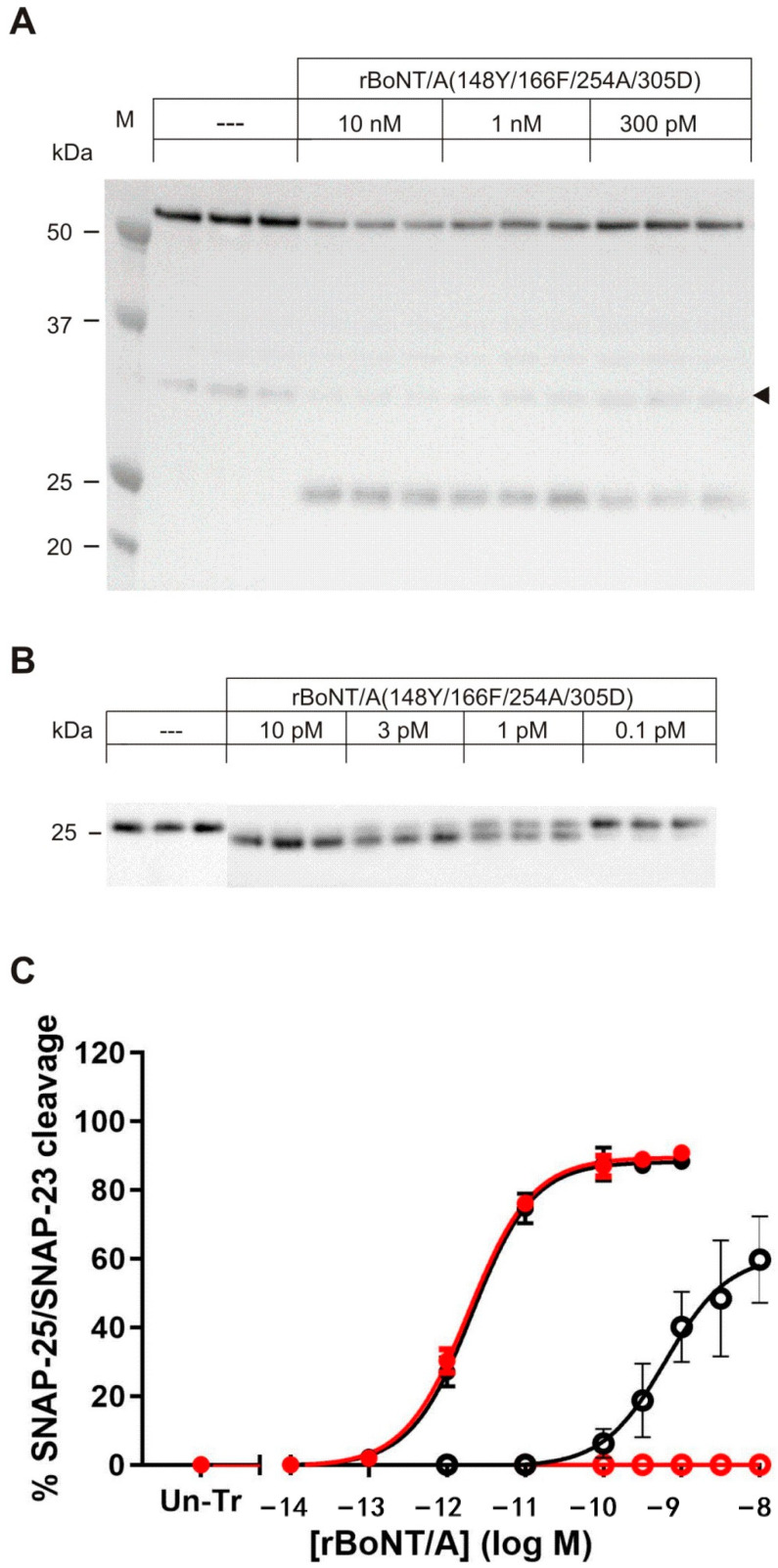
Activity of purified BoNT/A (E148Y, K166F, S254A, G305D) in cortical neurons. Cultivated rat cortical neurons were transduced with lentiviral vector expressing a hSNAP-23-mCherry fusion protein at DIV6. At DIV18 wild-type and virally transduced cells were exposed to varying concentrations of recombinant full-length wild-type or quadruple mutant of BoNT/A. After 24 h of incubation cells were lysed, and lysates analyzed for SNAP-25 and hSNAP-23 by specific antisera in Western blots. (**A**) Detection of hSNAP-23 using a rat polyclonal antiserum to human SNAP-23. Uncleaved hSNAP-23-mCherry migrates at about 55 kDa, whereas the presumed hSNAP-23 cleavage product appears at 25 kDa. The arrowhead marks an unidentified product. (**B**) Detection of endogenously expressed SNAP-25 using an antibody to the N-terminus of SNAP-25. The motility of the cleavage product is slightly faster due to loss of nine C-terminal amino acid residues. (**C**) Comparison of the activity of recombinant BoNT/A wild-type and quadruple mutant. Several analyses (four for quadruple mutant; five for wild-type) conducted like those shown in A and B were quantified and depicted as a function of BoNT concentration. SNARE cleavage was determined for wild-type (red) quadruple mutant (black) with SNAP-25 cleavage shown by filled circles and hSNAP-23 shown by open circles. Calculated EC_50_ values are shown in Appendix A.

**Table 1 toxins-12-00804-t001:** Activity of LC/A mutants to hSNAP-23 in standard in vitro cleavage assays.

	% Cleavage ± SD			
LC/A	30 nM	100 nM	1 µM	3 µM
wild-type	n. d.	n. d.	0.4 ± 0.5 ^a^	6.2 ± 0.7 ^a^
E148Y	n. d.	n. d.	9.7 ± 3.0 (15)	22.8 ± 4.7 (15) ^b^
K166F	n. d.	15.2 ± 3.1 (5)	66.1 ± 4.7 (5)	92.2 ± 2.5 (5)
K166I	n. d.	n. d.	9.8 ± 1.0 (4)	19.5 ± 3.3 (4)
K166L	n. d.	n. d.	42.7 ± 4.0 (4)	65.2 ± 6.0 (4)
K166V	n. d.	n. d.	9.6 ± 2.4 (4)	16.9 ± 4.3 (4)
S254A	n. d.	n. d.	9.3 ± 1.3 (2)	23.4 ± 0 (2)
S254R	n. d.	n. d.	4.0 ± 0.2 (4)	8.2 ± 0.8 (4)
E148Y/G305D	n. d.	n. d.	25.4 ± 2.3 (3)	53.0 ± 12.2 (3)
K166F/G305D	9.8 ± 6.3 (2)	23.5 ± 6.9 (4)	n. d.	n. d.
E148Y/K166F	18.9 ± 6.9 (7)	45.4 ± 10.9 (9)	n. d.	n. d.
E148Y/K166F/G305D	35.8 ± 6.2 (5)	79.0 ± 1.5 (3)	n. d.	n. d.
E148Y/K166F/S254A	37.4 ± 13.2 (6)	77.0 ± 11.4 (6)	n. d.	n. d.
E148Y/K166F/ S254A/G305D	64.1 ± 8.4 (6)	93.6 ± 0.6 (2)	n. d.	n. d.

^a^ data collected under identical experimental conditions were taken from reference [18]. ^b^ numbers in parenthesis specify number of experiments. n.d. not determined

**Table 2 toxins-12-00804-t002:** Activity of LC/A mutants to hSNAP-23 in simplified in vitro cleavage assays.

LC/A	% Cleavage ^a^	SD	No. of Exp.	Relative Activity
wild-type	17.7	5.2	22	1
S143D	36.9	3.5	8	2.09
S143E	43.3	4.9	6	2.45
S143Q	28.9	15.6	2	1.64
V304D	40.7	4.5	6	2.30
V304E	32.1	2.2	4	1.82
V304Q	27.0	3.0	6	1.53
G305D	34.7	3.2	6	1.97
G305E	25.7	3.1	4	1.45
G305Q	17.3	1.5	4	0.98
S143Q/V304D	30.5	4.2	4	1.73
S143Q/G305D	41.0	3.1	4	2.32
S143Q/G305E	29.1	3.1	4	1.65
S143Q/G305Q	29.4	3.2	4	1.66

Values represent means ± standard deviation. ^a^ cleavage assays were conducted at a 10 µM LC concentration.

**Table 3 toxins-12-00804-t003:** Enzyme kinetic parameters of wild-type LC/A and various mutants for hSNAP-23 cleavage.

LC/A	*K* _m_ _[µM]_	*k* _cat_ _[1/min]_	*k* _cat_ */K* _m_ _[1/min × µM]_	Relative Activity
wild-type	225 ± 38.2 (8) ^a^	0.20 ± 0.07	0.89 × 10^−3^	1
E148Y	51.8 ± 5.6 (2)	0.37 ± 0.18	7.1 × 10^−3^	8.0
K166F	44.0 ± 8.8 (4)	0.90 ± 0.10	20.4 × 10^−3^	23.0
V304D	56.1 ± 13.4 (3)	0.14 ± 0.07	2.45 × 10^−3^	2.8
V304E	65.5 ± 0.1 (2)	0.16 ± 0.04	2.4 × 10^−3^	2.7
G305D	52.0 ± 6.5 (2)	0.13 ± 0.02	2.5 × 10^−3^	2.8
G305E	61.5 ± 12.0 (3)	0.12 ± 0.003	1.9 × 10^−3^	2.2
S143D	68.9 ± 13.8 (3)	0.14 ± 0.03	2.0 × 10^−3^	2.3
S143E	57.1 ± 11.4 (2)	0.14 ± 0.02	2.3 × 10^−3^	2.6
S143Q	91.2 ± 12.3 (2)	0.21 ± 0.035	2.3 × 10^−3^	2.6
E148Y/K166F	16.9 ± 2.9 (4)	3.3 ± 1.0	195.3 × 10^−3^	220
E148Y/G305D	24.2(1)	0.29	12.0 × 10^−3^	13.5
E148Y/K166F/ G305D	24.3 ± 3.9 (4)	17.7 ± 0.9	728.4 × 10^−3^	820
E148Y/K166F/ S254A	18.6 ± 2.3 (4)	13.8 ± 1.0	741.9 × 10^−3^	835
E148Y/K166F/ S254A/G305D	15.9 ± 3.2 (4)	27.8 ± 4.1	1.75	1969
wild-type ^b^	9.8 ± 3.1	1026 ± 424	104.7	117,898

Values represent means ± standard deviation from 3 or 4 independent experiments. ^a^ numbers in parenthesis specify number of experiments. ^b^ activity versus human SNAP-25; data collected under identical experimental conditions were taken from reference [22].

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
