# Peer review of "Engineering an Effective Human SNAP-23 Cleaving Botulinum Neurotoxin A Variant"

_toxins, 2020, doi:10.3390/toxins12120804_

Round 1

Reviewer 1 Report

This study was dedicated to improvement of LC/A proteolytic activity against hSNAP-23. The authors identified four mutations that increased the mentioned activity even to 23-fold. They noticed, by using neuronal and cell-based assay,s that individual combination of these mutations led to visible synergistic effect on proteolytic activity of LC/A.

BoNT/A establishes pharmaceutical for treatment of medical conditions linked with hyperactivity of cholinergic nerve terminals. Especially, ovesecretion in non-neuronal cells is considered to be the cause of various diseases, e.g. chronic pulmonary disease, asthma diabetes etc. The aplication of BoNT/A in mentioned conditions is hampered by SNAP-23 responsible for exocitosis and resistant to BoNT. The authors described LC/A quadruple mutant which has potential to form the basis for new molecules with potential therapeutic use in conditions where SNAP-23 oversecretion causes pathology.

This study is written in clear and with very readable language and highlight potential new therapeutic use of BoNT/A. In my opinion it is distinguishing with high novelty and oryginality, therefore I recommend to publish this manuscript in present form (with little editorial improvements, e.g. lack of italics somewhere, etc). 

Author Response

(1) Referee: “…I recommend to publish this manuscript in present form (with little editorial improvements, e.g. lack of italics somewhere, etc.)”

Response: We have converted the terms “E. coli” and “in vitro” to italics throughout the text.

Reviewer 2 Report

The manuscript ‘Engineering an effective human SNAP-23 cleaving botulinum neurotoxin A variant’ presents an interesting study which follow the previous work done by the authors (reference 18 in the text).

However, the present manuscript is quite difficult to read and understand, without a precise knowledge of this previous work.

It should be clearly mentioned that the model of the complex between LC/A and SNAP-23 has been previously build, and that the present work is based on this model, for example at the paragraph beginning lines 108, and beginning line 164. I have only understood line 309 that the model has been previously build by the authors of this work.

It is also true in the abstract. It is written ‘based on the structure data and mutagenesis studies on SNAP-23’, I thus understand that this work was based on structural data from SNAP-23. It should me clearly stated that this work is based on structural data from a complex between LC/A and SNAP-25, or a structural model of a complex between LC/A and SNAP-25, together with mutagenesis studies on SNAP-23.

Line 113: the list of LC/A residues which bind SNAP-23 or SNAP-25 are clustered in two areas. It would be easier to understand if they are at that stage listed in two groups, one group interacting with P182 (R176) and one group interacting with K206 (T200).

Line 156: It is not clear to understand why mutants 9 and 10 have been chosen. Please be more explicit.

Line 169:  ‘dynamic structural modeling results’: what does that mean?

Line 181: No mention in the text of the mutants S254A and S254R, there are only mentioned in the tables.

In conclusion, modifications in the text which could facilitate the understanding of this interesting work would be useful.

Minor points

Line 12 : no previous definition of the abbreviation LC/A

Line 146 : Tab. 2 instead of Tab. 3

Fig. 1E: no mutant on this figure, while the figures B, C and D present some mutants. Why?

.

Reviewer 3 Report

This study is evaluating the engineering of a SNAP-23 cleaving BoNT/A variant.  The abstract, introduction, and discussion are very well written, although the authors seem to avoid commas throughout the entire manuscript. However, the results section is dense and difficult to follow, and the materials and methods section is lacking important details.  Statistical analysis is not included, even though it appears that experiments were conducted to enable appropriate statistical analyses.  The following detailed comments outline the major points that should be addressed.

Line 91:  replace ‘worse’ with ‘lower’ or another more descriptive term

Line 92:  it is a bit confusing to start results with Table 3, and only find the relevant data from the large table that is showing much more than the results described here.  Please consider describing the kcat in the text (0.2 vs 424) or creating a separate small table for this. 

Line 97:  replace brackets with parenthesis. 

Table 1:  Unless reference 18 is from the same authors using the same assay and reproducibility has been shown, it should not be used for comparison here.  Please specify in the comment under the table.

Table 2:  % wt is confusing.  Please choose one format to present data, either as the % cleavage or the  adjusted % cleavage versus wt cleavage.

Tables 1-3:  Please add statistical significance

Lines 115/116:  Even though the yeast screening tool is cited (although not described by name), it would be very helpful to include a short description here or in the materials section, at least describing the general mechanism of the screen.  The results presented here cannot be understood without looking up the cited papers.

Line 131:  ‘in vitro cleavage assays’.  Please be more specific and define exactly which assay(s) were used for this.  There’s only one cleavage assay and one enzyme kinetics assay described in the methods.

Line 133:  please specify increases in what?

Lines 151/152:  ‘seven, eleven, seven were counted …’  Please explain

Line 167:  ‘As the Km value of LC/A for interaction ….”.  In line 92 you mention a 5,000 fold lower kcat as particularly important in the difference in SNAP-25 vs SNAP-23 cleavage. 

Line 185:  ‘in vitro cleavage assays’.  Same comment as for line 131.  It would really be helpful for the reader to be presented with the basics of the type of assay, i.e. full length SNAP-23 and full length SNAP-25, recombinant LC 1-425.  In addition, I would like to see the full length proteins, as it is not easy to express these proteins full length in stable and soluble form.  Please also explain why beta-mercapto-ethanol was added and how that didn’t interfere with the endopeptidase assay.  The assay should be described such that it can be reproduced by others, and the details enabling such reproduction are lacking here.

Line 193:  please change effecting to resulting in

Lines 210-212:  Please specify how you tested the mutants.

Line 223/224:  Again, please specify the test used.  A statement like using the same endopeptidase assay with varying concentrations of substrate would suffice.

Line 239:  Again, please specify the assay used including the substrate (full length, his tagged) and read-out.  Was only a single concentration of substrate and of enzyme used? 

Line 240:  ‘cleavage rate’.  How would this endopeptidase assay determine a cleavage rate? 

Suppl table 3:  The % of wt LC A column is confusing, please replace with a more descriptive header (I assume it is  % cleavage by mutant vs % cleavage by wt LC).  It actually would be preferred to show that data in one format, either % cleavage or % cleavage with wt adjusted to 100%.  In addition, please indicate the n number and statistical significance.

Fig 1:  The aa numbers are a bit hard to read, especially in (a).  Is it possible to change colors of the numbers to black, or bold them?

Line 261:  While cortical neurons do not express SNAP-23, it is almost impossible to achieve a primary cortical neurons culture completely devoid of glial cells, which would express SNAP-23.  Did the authors examine the cell culture for SNAP-23 expression?  I’m amazed that there’s no apparent interference on the Western blot from endogenously expressed SNAP-23.

Lines 273-275:  You list single EC50 values here, but the suppl figure says the experiment was conducted in triplicate.  Please list average and standard deviation of EC50s and a statistical analysis, which should dhow no difference.  ‘virtually identical’ is not a scientific term and is vague.

Lines 278-280”  same comment as above.  Please include a statistical analysis, and a range for the fold difference or CI.

Fig 2B:  The marker lane is rather confusing here and not at all useful, as it shows just one band.  Please remove.

Suppl Table 4 and 5 should include 95% CI

Lines 384-386:  This last sentence really summarizes the most important aspect of this, yet places no emphasis on the enormous challenge that targeting the desired cells and avoiding neurons pose. 

Lines 455 ff:  Please add the buffer in which the cleavage assay was conducted.

Line 469:  please define toxin assay buffer again, even though it is defined in the previous section.

Lines 497/498:  Please describe the lentivirus trsnduction in more detail.  Was it done in culture media, PBS, or another buffer?  For how long?  Temp?  etc.  This is not an easy system to set up and again, the methods description should enable replication of the experiment.
In addition, please add images showing successfully transduced cells, which will show what percentage of cells was transduced. 

Some general comments:

Materials and Methods:  Please add a section on statistical analyses including n number etc.  The data are shown as presumably average or means and standard deviation, but it is unclear how these values were derived.  Only table 3 specifies this.

Results section 2.1:  This section is a bit unclear and it is quite dense and difficult to read.  I didn’t fully understand it until I read the discussion section.  Please explain the approach and thinking a bit more clearly in this section, the results should be understandable on their own.

Why not cite the previous study that created a SNAP-23 cleaving BoNT?  Engineering botulinum neurotoxin to extend therapeutic intervention.  Sheng Chen 1, Joseph T Barbieri
